# Chronic Rupture of Achilles Tendon Caused by Haglund’s Deformity: A Case Report

**DOI:** 10.3390/medicina58091216

**Published:** 2022-09-04

**Authors:** Muhammad Andry Usman, Benny Murtaza, Putu Acarya Nugraha Winangun, Dave Kennedy

**Affiliations:** Department of Orthopedics and Traumatology, Faculty of Medicine, Hasanuddin University, Makassar 90245, Indonesia

**Keywords:** Achilles tendon, case report, flexor hallucis longus tendon transfer, Haglund’s deformity, tendon rupture

## Abstract

A chronic Achilles tendon rupture is a tendon rupture occurring more than 4–6 weeks after a traumatic injury. Haglund’s deformity, caused by bony abnormalities in the ankle (mostly due to osteophyte or bone spur), can cause chronic inflammation and degeneration of the Achilles tendon, eventually leading to rupture. This presents a challenge for clinicians who provide tendon repair procedures. We present a 69-year-old woman who had difficulty moving her left leg and had a deformity on the left leg compared to her right leg after falling nine months before but with pain starting three months before the accident. There was a seven-centimeter gap in the calcaneus with a positive Thompson test. The Haglund’s deformity on the left calcaneus was visible on the ankle X-ray. The patient had a chronic total rupture of the left Achilles tendon, which was treated with a flexor hallucis longus (FHL) tendon transfer and resection of the deformity. One week after surgery, the patient’s ability to walk and the shape of the left leg improved. This case report describes a chronic left Achilles tendon condition that was successfully repaired through tendon repair surgery using FHL tendon transfer and removal of Haglund’s deformity.

## 1. Introduction

The Achilles tendon is considered the largest and strongest tendon in the human body. Nonetheless, this structure is the most frequently ruptured tendon in the lower limbs, particularly due to degenerative (25%) and mechanical/sport-related (75%) processes [1,2]. Several characteristics were discovered to be significantly distinct between patients with tendinopathy and the normal population, including age; body mass index (BMI); abductor hallucis brevis muscle thickness; extensor digitorum longus muscle cross-sectional area (CSA); and both the thickness and CSA of the tibialis anterior, flexor digitorum brevis, flexor hallucis brevis, and peroneus muscles [3,4].

In elderly patients, a bony spur (osteophyte) may form at the calcaneus, causing some disruptions to the Achilles tendon attached to it. Haglund’s deformity (an abnormality of the bone and soft tissues in the foot) may develop later in life due to an expansion of the bony section of the heel [5]. The deformity can be detected by clinical (mainly inspection and palpation) and radiographical examinations [6]. Open surgery (bursectomy and calcaneal exostosis resection) and endoscopic calcaneoplasty are two treatment options available for this disorder [7]. Haglund’s deformity was found to be responsible for acute Achilles tendon rupture, particularly in athletes [6]. However, its importance has not been addressed adequately in chronic Achilles tendon rupture cases.

Early detection and treatment should be carried out properly for Achilles tendon ruptures. However, it is estimated that about 25% of Achilles tendon injuries are not detected early, resulting in a neglected condition known as chronic Achilles tendon rupture (delayed diagnosis and procedure for more than 4–6 weeks) [8]. It is primarily attributed to underdiagnosis, with the injury referred to as hematoma formation or simply an ankle sprain, especially when the patient can walk to the examination room [9,10]. Chronic Achilles tendon rupture necessitates more difficult treatment due to the retraction of the tendon ends and the surrounding muscles, scar tissue development in the gap, and diminished gastrocnemius muscle contractility [11,12,13]. Flexor hallucis longus (FHL) tendon transfer is a renowned and minimally invasive surgery method for treating chronic Achilles tendon rupture cases. Some studies have reported favorable outcomes following the restoration of the ruptured Achilles tendon with the FHL tendon transfer procedure. Furthermore, this technique can preserve the integrity of the skin along the affected region [14,15].

Considering the evidence stated earlier in this section, chronic Achilles tendon rupture presents a great challenge, particularly for clinicians performing surgical procedures on that tendon. In addition, osteophyte formation may be the basis of Haglund’s deformity, one factor that can trigger Achilles tendon rupture. The combination of traumatic events affecting the Achilles tendon and this deformity on the calcaneal bone is still scarcely studied. As a result, the authors present a case report of a chronic total rupture of the left Achilles tendon in an elderly woman caused by Haglund’s deformity that was surgically repaired using FHL tendon transfer.

## 2. Clinical Case

A 69-year-old female reported the inability to move her left ankle properly for about one year, which was accompanied by heel pain. The patient also reported a fall after slipping with her legs bent inward (inversion position) about three months afterward. Following the accident, the patient experienced worsening pain with every movement in her left ankle, causing difficulty walking. Furthermore, the patient’s relative also noticed differences in the shape of the left ankle compared with the right ankle, which was primarily observed when praying. She did not seek any treatment during this period; hence, no improvement in her complaints was witnessed. Afterward, she went to a Regional General Hospital, was diagnosed with Achilles tendon rupture, and was then referred to our hospital.

The inspection of the left ankle revealed the presence of Haglund’s deformity and a positive Matles test (the absence of plantar flexion). The deformity could be palpated as a bony prominence on the posterosuperior part of the calcaneus. No swelling, hematomas, or wounds were observed in the region (Figure 1). There was no pressure pain in the lateral part of the knee joint on palpation, but there was a gap formation, stretched about seven centimeters between the calcaneus bone and the Achilles tendon. The movement of her left ankle triggered pain (NRS: 3/10). Active movement to the dorsiflexion, eversion, and inversion positions could be performed adequately, but plantarflexion was not observed in the patient’s left ankle. Passive movement yielded a normal range of motion (ROM). The Thompson test was positive on the left ankle. A plain ankle radiograph of the lateral projection of the patient’s left ankle revealed the presence of osteophytes in the calcaneus bone and confirmed Haglund’s deformity (Figure 2).

The patient was diagnosed with a chronic total rupture of the left Achilles tendon, which was hypothesized to be caused by the formation of Haglund’s deformity. Surgery was conducted in a prone position. A skin incision was created at the medial rim of the Achilles tendon and then explored to the deep fascia until the Achilles tendon injury was viewable and the pointed edge of Haglund’s deformity was revealed. The Haglund’s deformity was resected, and the osteophyte was removed via surgical exostectomy. Tendon repair surgery was selected as the treatment modality using flexor hallucis longus (FHL) tendon transfer in conjunction with osteophyte removal. FHL tendon transfer was performed under general anesthesia. Briefly, the patient was situated in a prone position, followed by several stages: debridement (oriented to remove the fibrotic scar at the end of the tendon), identification (the exploration process to precisely identify the structure of the Achilles tendon and FHL), measurement of the FHL tendon, and harvesting (isolation of the FHL tendon). Finally, augmentation and fixation steps were performed to strengthen the Achilles tendon’s bond and settle it alongside the FHL tendon. (Figure 3).

Following the completion of the tendon reconstruction, the wound was closed with a compressive dressing and a splint was used to maintain the ankle position in 150° plantarflexion (Figure 4). There were no postoperative complications or signs of infection in the surgical wound, with mild pain reported by the patient (NRS: 1/10). The patient was advised to walk on crutches for 4–6 weeks while wearing slabs. Non-weight-bearing exercises were performed for one month and were then shifted to weight-bearing exercises the following month. After the one-week follow-up, the patient still had a limited active and passive range of motion (ROM). One month after the surgery, the patient reported minimal pain with limited (but improved) ROM (from grade 0 to grade 2) of her left ankle. She is still receiving physiotherapy on a regular basis to train the movement of her left ankle. An improvement was noted in the shape of the patient’s left leg, which was no longer significantly different from her right leg (Figure 5). The patient consented to the publication of this case in the journal.

## 3. Discussion

Several studies have shown that tendon rupture caused by degenerative processes primarily affects the elderly population. A rupture can be mediated by a minor injury, such as a fall or slip [16] This is due to microstructural changes related to collagenization, vascularization, and hypercellularization, which make the Achilles tendon prone to spontaneous rupture. In our patients, the onset of the disease is about one year, supporting the diagnosis of chronic Achilles tendon rupture [11,17,18].

The measurement of the gap in the chronic rupture of the Achilles tendon can help to determine the functional prognosis of the patient. In one study, a gap of >5 mm in the tendon provided a better prognosis in patients with low physical demand than a gap measured at 10 mm. Thus, surgery can provide good results on patient function [19,20].

Calcaneal (heel) spur is a condition in which bone grows above the calcaneus. It can grow on the dorsal or the plantar side of the ankle. Dorsal spurs are frequently associated with insertional Achilles tendinopathy, caused by a degenerative process (osteophyte formation) in the patient’s ankle [21]. An osteophyte can trigger a constant pull effect, which causes a chronic inflammatory reaction. This inflammatory response causes changes in the structure of the Achilles tendon, making it to be stiffened and less easily stretched, possibly leading to a rupture even with minor injury [22]. A tendon rupture following tendinopathy is estimated to occur in 4% of individuals and is mainly attributed to the Achilles tendon’s intrasubstance degeneration [17]. The presented patient was diagnosed with a chronic total rupture of the left Achilles tendon [23,24]

The degenerative process is also triggered by mechanical friction caused by Haglund’s deformity, resulting in irritation of the neighboring soft tissues and insertional Achilles tendinitis [6]. A chronic inflammatory process reduces Achilles tendon strength and can lead to tendon rupture, even in the presence of minor trauma. Clinical examination and radiographic findings are commonly employed to diagnose Haglund’s deformity [25].

Reconstructive surgery utilizing tendon transfer can help restore the Achilles tendon’s strength, improving the patient’s quality of life. The FHL tendon is frequently used for tendon transfer techniques. It is the closest tendon to the Achilles tendon [26,27]. The surgical resection of the spur and Haglund’s deformity can be performed in patients with evidence of calcaneal spur development. This can lead to the decompression of the insertion of the Achilles tendon. The posterior calcaneus can then be smoothed to prevent any remaining irritation between the bony edges and the tendon [28]. The combination of FHL tendon transfer and Haglund’s deformity removal can be beneficial to the patient’s outcome.

FHL tendon transfer is accomplished by harvesting the tendon and transferring it to the bone canal on the posterior calcaneus. This technique is helpful for Achilles tendon reconstruction because of the characteristics of the FHL tendon, one of the longest tendons with similar actions to the Achilles tendon. In addition, the FHL tendon provides adequate strength to assist the Achilles tendon’s function [27]. The FHL tendon can be used to bridge the gap created by the total rupture of the Achilles tendon, which is supported by the plantarflexion leg position. The FHL technique is regarded as the most theoretically advantageous method of tendon repair [19,29].

The patient was scheduled to wear a four-week non-weight-bearing cast on the leg postoperatively. Afterward, the patient was given a short-leg walking cast or a removable cast walker to maintain the ankle in a neutral position for four weeks. Then, weight-bearing exercises were prescribed to the patient. Physiotherapy rehabilitation in patients following Achilles tendon repair typically begins in the eighth week following surgery. Athletic activity was also restricted for six weeks following surgery [30]. A week after surgery, the patient could already walk. Following her surgical procedure, the patient showed improved ROM, functional status, and pain status of the ankle. Patient-reported outcomes have been more frequently used during the past decade to obtain the patient’s own opinions about their results; thus, the practitioner can obtain an overall picture of the treatment results [31,32].

## 4. Conclusions

This case report describes an elderly patient with a chronic total rupture of the left Achilles tendon resulting from Haglund’s deformity of the calcaneus. The deformity was responsible for the degenerative process and injury to the surrounding tissues. FHL tendon transfer and surgical resection of Haglund’s deformity and the osteophyte were used to resolve the rupture. Chronic Achilles tendon rupture is more difficult to manage than acute Achilles tendon rupture because it is associated with scar tissue formation at the tendon’s gap, causing ankle dysfunction. FHL tendon transfer improved the patient’s clinical condition, as evidenced by functional improvement (improved movement), anatomical improvement (improvement of the difference in leg shape), and pain amelioration.

## Figures and Tables

**Figure 1 medicina-58-01216-f001:**
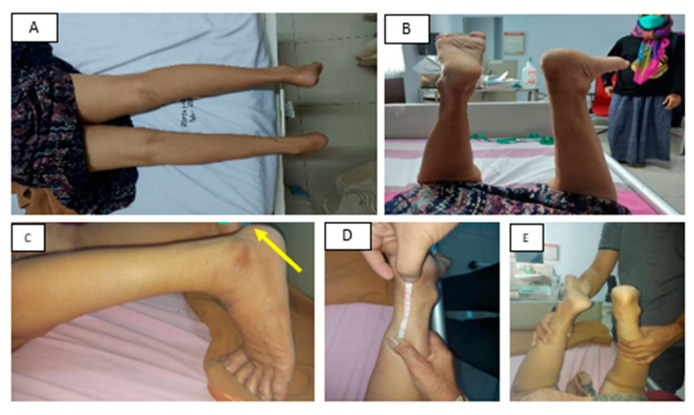
Preoperative clinical photo. (**A**,**B**) clinical picture of the patient’s legs; (**C**) Haglund’s deformity (arrow); (**D**) the measurement of the seven-centimeter gap between the insertion of the Achilles tendon and the calcaneus bone; (**E**) positive Thompson test.

**Figure 2 medicina-58-01216-f002:**
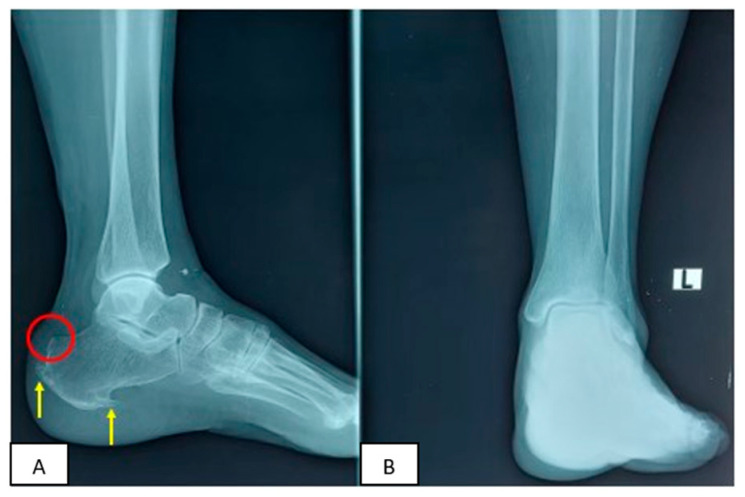
Preoperative X-ray of left ankle: (**A**) lateral view; (**B**) anteroposterior view. Note the observed prominent Haglund’s deformity (red circle) with dorsal and plantar osteophyte formation on the calcaneus bone (arrow).

**Figure 3 medicina-58-01216-f003:**
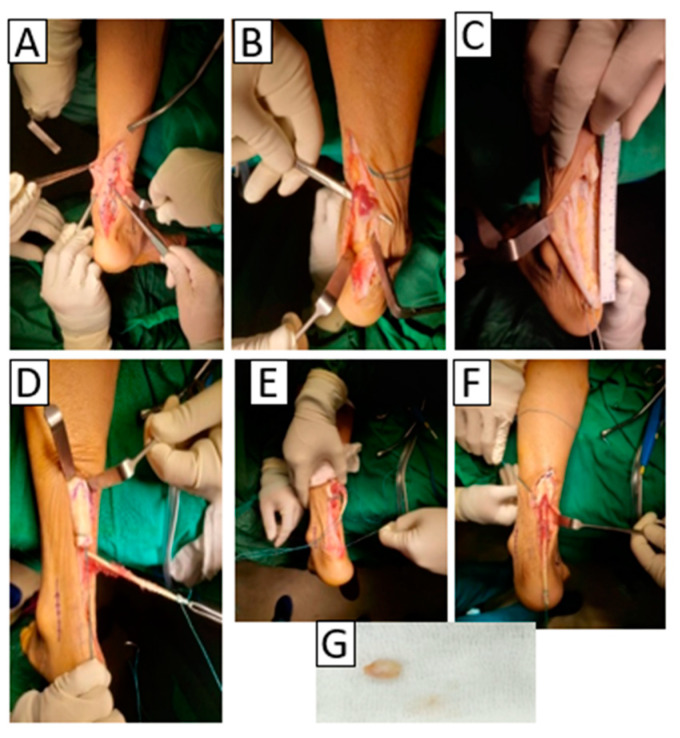
FHL tendon transfer and calcaneal spur reduction. (**A**) Debridement and Haglund’s deformity detection, (**B**) identification, (**C**) FHL tendon measurement, (**D**) harvesting, (**E**) augmentation, (**F**) fixation procedures, (**G**) extracted calcaneal spur.

**Figure 4 medicina-58-01216-f004:**
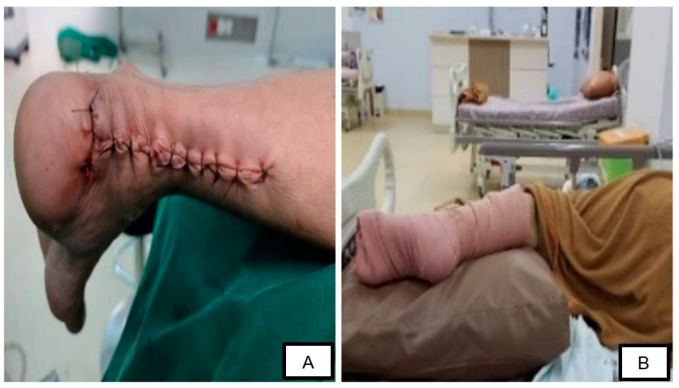
Postoperative clinical condition: (**A**) surgical wound closure, (**B**) fixation at 150° plantarflexion.

**Figure 5 medicina-58-01216-f005:**
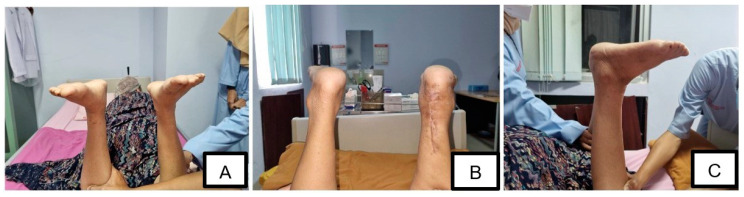
Follow-up condition (one month after the surgery). (**A**) anterior view, (**B**) posterior view, (**C**) lateral view. Note that plantarflexion can be noticed in both limbs (negative Matles test).

## Data Availability

Not applicable.

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
