# Peer review of "Chronic Rupture of Achilles Tendon Caused by Haglund’s Deformity: A Case Report"

_medicina, 2022, doi:10.3390/medicina58091216_

Round 1

Reviewer 1 Report (Previous Reviewer 2)

I appreciated the changes made to the original manuscript: these changes made the topic more understandable and the manuscript more complete. 

Reviewer 2 Report (Previous Reviewer 3)

I thank the corresponding author for their comments. I have read through the subsequent changes made to the manuscript and I have no further comments or suggestions.

This manuscript is a resubmission of an earlier submission. The following is a list of the peer review reports and author responses from that submission.

Round 1

Reviewer 1 Report

Thank you for the effort you put into the article.

The relationship of the cause and the conclusion  you have stated in the article is not clear.

Findings on radiography (presence of osteophytes)  are common and do not support the hypothesis

Author Response

EXPLANANATION OF STATEMENT, COMMENT AND RESULT OF REVISED PAPER FROM

REVIEWER 1

The highest thanks to reviewer who have patiently and carefully examined and revised our article titled: Chronic Rupture of Achilles Tendon Due to Osteophyte: A Case Report

I am very happy to receive your comment that you have presented in our article. I realize that there is a shortcoming in our article based on your review. Finally, I could construct better article with your substantially important issue addressed to us.

Thank you for the effort you put into the article.

  1. The relationship of the cause and the conclusion you have stated in the article is not clear.

Adjustment: Thank you for your comment. We have added the explanation of the phenomena called as Achilles tendon rupture following Achilles tendinopathy as the underlying process of the case reporting.

  1. Findings on radiography (presence of osteophytes) are common and do not support the hypothesis.

Adjustment: Thank you for your comment. We have added the explanation of the importance of osteophyte finding and its impact to the patient, especially with its location in the dorsal side of the calcaneus bone.

Reviewer 2 Report

The article is very well written and of clinical importance. I have only two suggestions for Authors: in the Introduction section, I suggest to add the reason why approximately 25% of Achille Tendon injuries are not detected early. And in the discussion section, I suggest to add other references, if present in literature. 

Author Response

EXPLANANATION OF STATEMENT, COMMENT AND RESULT OF REVISED PAPER FROM

REVIEWER 2

The highest thanks to reviewer who have patiently and carefully examined and revised our article titled: Chronic Rupture of Achilles Tendon Due to Osteophyte: A Case Report

I am very happy to receive your comment that you have presented in our article. I realize that there is a shortcoming in our article based on your review. Finally, I could construct better article with your substantially important issue addressed to us.

The article is very well written and of clinical importance. I have only two suggestions for Authors:

  1. In the Introduction section, I suggest to add the reason why approximately 25% of Achille Tendon injuries are not detected early.

Adjustment: Thank you for your comment. We have added the cause as an underdiagnosis process, when the rupture is regarded as the hematoma formation or just an ankle sprain, mainly while the patient is still able to walk to the examination room.

  1. And in the discussion section, I suggest to add other references, if present in literature.

Adjustment: Thank you for your comment. We have added more literature on it and restructure the discussion.

Reviewer 3 Report

Comments and Suggestions for Authors

Originality: This is a very short paper, providing a minimum of information about a case study, and constitutes a very minor contribution to the literature. The introduction section did not provide a clear rationale for carrying out the study (for example, why is your research question important? What gap in the literature is the study addressing?). I suggest to describe in this section only with the information related with the state of art related with the chronic Achilles tendon factors, aetiology and treatment.

Thus, I think in this section should be improved, with more details about the  patients with Achilles tendinopathy https://pubmed.ncbi.nlm.nih.gov/31593918/. Furthemore, to revise the researches of Romero Morales et al related with xtrinsic foot muscles in patients with chronic non-insertional Achilles tendinopathy https://pubmed.ncbi.nlm.nih.gov/30844628/

Methodologically Sound: As a case study report it is rather hard to go wrong methodologically, and the paper conforms to the standard.

Follows Appropriate Ethical Guidelines: Whilst there is no obvious declaration of ethical approval. Please include the date and code register number of ethics committee it would appear to be a report of actions taken as part of normal clinical practice (as a case study report), and thus is acceptable.

Has results which are clearly presented and support the conclusions: Again, it conforms to the usual format for the presentation of a case study, although the content is very sparce.  It is, however, appropriate enough, and does report a rare case likely to be of interest to a healthcare audience.  

Overall Scientific Quality:  As a minor case study report it lacks scientific depth, but effectively is intended only to report the occurrence of a typical case and to highlight the importance of correct diagnosis, and on these grounds merits attention.

Presentation, Organization, Clarity:  I think you have some good information. But it is poorly presented. You will have to totally rewrite the manuscript.

Incorrectly References Previous Relevant Work:  It not appears to reference prior work succinctly and accurately.

Importance/Interest: Although marked by its brevity, the content is of interest, particularly to clinicians such as traumatologist who examine chronic Achilles tendon who may need to be aware of the variant forms of this illness.

Author Response

EXPLANANATION OF STATEMENT, COMMENT AND RESULT OF REVISED PAPER FROM

REVIEWER 3

The highest thanks to reviewer who have patiently and carefully examined and revised our article titled: Chronic Rupture of Achilles Tendon Due to Osteophyte: A Case Report

I am very happy to receive your comment that you have presented in our article. I realize that there is a shortcoming in our article based on your review. Finally, I could construct better article with your substantially important issue addressed to us.

  1. Originality: This is a very short paper, providing a minimum of information about a case study, and constitutes a very minor contribution to the literature. The introduction section did not provide a clear rationale for carrying out the study (for example, why is your research question important? What gap in the literature is the study addressing?). I suggest to describe in this section only with the information related with the state of art related with the chronic Achilles tendon factors, aetiology and treatment. Thus, I think in this section should be improved, with more details about the patients with Achilles tendinopathy https://pubmed.ncbi.nlm.nih.gov/31593918/. Furthemore, to revise the researches of Romero Morales et al related with xtrinsic foot muscles in patients with chronic non-insertional Achilles tendinopathy https://pubmed.ncbi.nlm.nih.gov/30844628/

Adjustment: Thank you for your comment. We have revised the introduction as your guidance. We also added the suggested reference. In addition, we have tried to present the gap and importance in the introduction section.

  1. Methodologically Sound: As a case study report it is rather hard to go wrong methodologically, and the paper conforms to the standard.

Adjustment: Thank you for your comment.

  1. Follows Appropriate Ethical Guidelines: Whilst there is no obvious declaration of ethical approval. Please include the date and code register number of ethics committee it would appear to be a report of actions taken as part of normal clinical practice (as a case study report), and thus is acceptable.

Adjustment: Thank you for the comment. We agree with your input on the guideline for an ethical availability of the case report, which is still not obvious. However, we also provided the ethical approval in the paper (protocol code 10903/UN4.5/KEP-FKUNHAS/2021, 16th December 2021) below the conclusion section, as per the journal guideline.

  1. Has results which are clearly presented and support the conclusions: Again, it conforms to the usual format for the presentation of a case study, although the content is very sparce. It is, however, appropriate enough, and does report a rare case likely to be of interest to a healthcare audience. 

Adjustment: Thank you for your comment.

  1. Overall Scientific Quality: As a minor case study report it lacks scientific depth, but effectively is intended only to report the occurrence of a typical case and to highlight the importance of correct diagnosis, and on these grounds merits attention.

Adjustment: Thank you for your comment.

  1. Presentation, Organization, Clarity: I think you have some good information. But it is poorly presented. You will have to totally rewrite the manuscript.

Adjustment: Thank you for your comment. We have revised the manuscript flow according to your input and other reviewer’s input. The manuscript has been re-written with extensive editing.

  1. Incorrectly References Previous Relevant Work: It not appears to reference prior work succinctly and accurately.

Adjustment: Thank you for your comment. We have edited the referencing with new references to support our study.

  1. Importance/Interest: Although marked by its brevity, the content is of interest, particularly to clinicians such as traumatologist who examine chronic Achilles tendon who may need to be aware of the variant forms of this illness.

Adjustment: Thank you for your comment.

Round 2

Reviewer 3 Report

I am happy with the paper as it stands. Congratulations.